# Comparison of Cardiovascular Risk and Events among Spanish Patients with and without Ocular Pseudoexfoliation

**DOI:** 10.3390/jcm11082153

**Published:** 2022-04-12

**Authors:** Nora Imaz Aristimuño, Iñaki Rodriguez Agirretxe, Ricardo San Vicente Blanco, Rafael Rotaeche Del Campo, Javier Mendicute Del Barrio

**Affiliations:** 1Department of Ophthalmology, Donostia Universitary Hospital, 20014 Donostia-San Sebastian, Spain; ira@icqo.org (I.R.A.); javier.mendicutedelbarrio@osakidetza.eus (J.M.D.B.); 2Primary Care Unit, Health Centre of Zumarraga, 20700 Zumarraga, Spain; ricardo.sanvicenteblanco@osakidetza.eus; 3Primary Care Unit, Health Centre of Altza, 20017 Donostia-San Sebastian, Spain; rafael.rotaechedelcampo@osakidetza.eus

**Keywords:** cardiovascular, pseudoexfoliation, cataract, hypertension, coronary, stroke, vascular, mortality

## Abstract

The purpose of this study was to calculate and compare individual cardiovascular risk (CVR) and the development of cardiovascular events and mortality in patients with and without ocular pseudoexfoliation (PEX). A cohort study was carried out to compare two groups of patients who underwent cataract surgery: patients with (*n* = 99) and without PEX (*n* = 239). The CVR factors were recorded for all the subjects, and CVR was calculated for each individual using ERICE risk assessment charts. After a six-year follow-up, the cardiovascular events and mortality rates were compared between the two groups. The mean CVR was 36.41% in the PEX group and 33.72% in the non-PEX group (*p* = 0.13). High blood pressure was detected in significantly more PEX patients (71.4%) than non-PEX patients (58.6%, *p* = 0.035), yet no differences were found in the other CVR factors. The prevalence of cardiovascular events in the PEX and non-PEX patients was 17.1% and 12.5%, respectively (*p* = 0.26), with 5% of patients in the PEX group and 5% in the non-PEX group suffering an acute coronary event (*p* = 0.9). Moreover, 12% of the PEX patients and 7.5% of the non-PEX patients experienced a stroke (*p* = 0.17), and the six-year mortality rate was 29.3% in the PEX group and 25.9% in the non-PEX group (*p* = 0.52). PEX was associated with high blood pressure in our cohort of patients, although cardiovascular risk was not shown to be higher in this group. In addition, patients with PEX did not experience more cardiovascular events or have a higher mortality rate than patients without PEX during the period studied.

## 1. Introduction

Pseudoexfoliation syndrome (PEX) was first described by John Lindberg (Finland) in 1917 [1,2]. It consists of the deposition of a whitish material on the anterior lens surface, which becomes visible when the pupil is fully dilated. PEX is a systemic disorder, and in addition to eye tissue, PEX depositions may also be evident on the skin, in visceral organs, and on the vascular wall [3,4]. PEX involves elastosis that is caused by the excessive synthesis and reduced degradation of elastic microfibrils, provoking alterations to vessel walls. These findings suggest that vascular dysregulation in PEX syndrome could contribute to or exacerbate arterial diseases, including cardiovascular diseases.

Cardiovascular disease is one of the main causes of morbidity and mortality in developed countries, the most common cause of which is atherosclerotic damage to a coronary or brain arteries. The main risk factors for atherosclerosis are an increase in low-density lipoprotein (LDL) in plasma, a reduction in high-density lipoprotein (HDL), smoking, high blood pressure (HBP), and diabetes mellitus [5]. Cardiovascular risk (CVR) prediction charts allow the risk of experiencing a cardiovascular event to be predicted accurately, which is important, as a series of coordinated actions can help prevent cardiovascular diseases, both at the population and individual level. Indeed, these actions aim to eliminate or minimize the impact of cardiovascular diseases and the disability associated with them [6].

Several CVR calculation charts have been approved for the Spanish population. ERICE is a new native CVR equation for the low-risk and ageing Mediterranean population in Spain that was published in 2015 [7] (Appendix A). The ERICE charts estimate the absolute 10-year CVR risk for a first fatal or non-fatal cardiovascular event, and the risk factors included are age, smoking, diabetes mellitus, systolic blood pressure and total cholesterol. Moreover, the ERICE data include sections for people aged 70–79 or over 80 years of age and take into account the use of drugs that modify specific risk factors (e.g., anti-HBP treatment).

The aim of the present study was to assess the relationship between ocular PEX and cardiovascular disease, assessing whether PEX is an independent risk factor for cardiovascular disease or if it is associated with classic CVR factors.

## 2. Materials and Methods

This study was carried out according to the guidelines of the Helsinki Declaration, and the project was approved by the Clinical Research Ethics Committee of the Donostia University Hospital (HUD) in May 2014. All the participants in this study provided their signed informed consent prior to their enrollment in the study.

### 2.1. Study Population and Cohorts

The study was carried out at the HUD in Donostia-San Sebastián (Spain), a public tertiary referral hospital serving a population of about 400,000 inhabitants. The study population included all patients who underwent cataract surgery during the period from June to December 2014 at the HUD. The study was carried out on two groups of patients, with or without ocular PEX. All patients were included in a post-operative evaluation carried out the day after the cataract intervention. The patients were recruited over 7 months, from June to December 2014.

### 2.2. PEX Diagnosis

Data on the presence or absence of PEX were obtained from each patient’s ophthalmological medical records. All the patients underwent a complete ophthalmological examination under pharmacological mydriasis prior to cataract surgery. If there was evidence or signs of PEX or PEX glaucoma in either or both eyes in the patient’s medical record, the patient was included in the PEX cohort. If there was no mention of such signs of PEX in the patient’s medical record, the patient was included in the non-PEX cohort.

### 2.3. Inclusion Criteria

The criteria for inclusion were patients over 50 years of age with a medical history that included records of their ophthalmological examination prior to cataract and their medical–surgical history according to the 9th revision (ICD-9) of the International Classification of Diseases code system, with laboratory tests and blood pressure readings over the last year, and treatments received to date.

### 2.4. Exclusion Criteria

Patients were excluded from the study if they had any history of previous ischemic heart disease in their medical records, a history of coronary heart disease (ICD-9 codes 410–414), or any evidence of cerebrovascular disease (ICD-9 codes 430–438) in their medical records.

### 2.5. Data Collection for CVR Calculation

The following variables were collected from all the patients in the two cohorts, obtained from their medical records: diagnosis of diabetes mellitus (prescription of anti-diabetic drugs or a fasting basal blood glucose ≥ 126 mg/dL); total cholesterol (mg/dL); HDL cholesterol (mg/dL); LDL cholesterol (mg/dL); systolic blood pressure (mm Hg); diastolic blood pressure (mm Hg); high blood pressure medication (diuretic medications, beta-blockers, alpha-beta blockers, angiotensin-converting enzyme inhibitors, angiotensin II antagonists, calcium channel blockers, alpha-1-blockers); HBP diagnosis (defined by the prescription of any anti-HBP drug); and existence of a smoking habit (a person was considered to be a smoker if they had smoked daily in the last month, regardless of the number of cigarettes). Based on these data, the individual’s CVR was calculated using the ERICE risk calculation charts. The proper ERICE chart was used for each patient (diabetic vs. non-diabetic, hypertensive vs. non-hypertensive). Cholesterol values were converted to mmol/L to calculate the CVR.

### 2.6. Patient Follow-Up

Six years after the start of the study (December 2020), the medical records of all the participants were reviewed. According to the ICD-9 code system, any ischemic heart disease event (ICD-9 code 410–414) or stroke (ICD-9 code 430–438) suffered by the patients in this period was recorded. In addition, patient mortality was recorded without specifying the cause and based on the information in each patient’s clinical records.

### 2.7. Statistical Analysis

Statistical analysis of the data was performed using the IBM SPSS Statistics for Windows software, version 23.0 (IBM Corp., Armonk, NY, USA). A descriptive analysis was performed by calculating the mean and standard deviation of the continuous variables, and the frequency and percentages were used to analyze the categorical variables. Student’s *t*-test was used to compare the means, and an Analysis of Variance and a Chi-squared test were used to compare the proportions. Pearson’s correlation coefficient was used to measure the correlation between the continuous variables. Statistical significance was set at *p* < 0.05.

## 3. Results

A total of 1426 cataract interventions were registered at the HUD in the period from June to December 2014, of which 108 corresponded to patients with PEX. Hence, the prevalence of PEX in this cohort was 7.57%. Among these patients, five (4.6%) had a history of cerebrovascular disease and four (3.7%) of ischemic heart disease. These patients were therefore excluded from the study. The baseline characteristics and outcomes of the whole population are shown in Table 1.

Taking into account the inclusion criteria, the final cohort included 338 patients, consisting of 99 patients (29.3%) with PEX syndrome (PEX cohort) and 239 (70.7%) without PEX syndrome (non-PEX cohort). The epidemiological characteristics of the sample, as well as their respective CVR factors, are described in Table 2.

### 3.1. Calculation of the CVR According to ERICE and a Comparison of the CVR Factors between PEX and Non-PEX Patients

When CVR was calculated using the ERICE charts, no statistical differences between PEX and non-PEX patients were detected (Table 3). When the individual distributions of CVR factors were evaluated in the PEX and non-PEX cohorts, as expected, there were no differences regarding age and sex in both cohorts (Table 2). However, there were significant differences in the levels of total cholesterol and HDL cholesterol between these two groups of patients, and the proportion of patients with HBP was significantly higher in the PEX cohort.

This association of PEX with HBP was analyzed in the overall sample (*n* = 338), dividing this cohort into the patients diagnosed with HBP and patients without HBP, and analyzing the presence of PEX in both groups (Table 4). The univariate analysis revealed that PEX was associated with HBP, as confirmed subsequently in a multivariate analysis, with PEX presenting an odds ratio (OR) of 1.79 for the presence of HBP (confidence interval 1.06–3.1).

### 3.2. Analysis of Cardiovascular Events and Mortality

During the six years of follow-up, 47 (13.4%) patients in the overall sample (*n* = 338) suffered a cardiovascular event, of which 17 patients suffered acute ischemic heart disease (5%), and 30 had an acute stroke (8.8%). The prevalence of a cardiovascular event in the PEX cohort was 17.1%, and it was 12.5% in the non-PEX cohort (*p* = 0.26, chi squared). Moreover, 5 PEX patients (5%) and 12 non-PEX patients (5%) suffered an acute coronary event (*p* = 0.9, chi squared). With regards to cerebrovascular disease, 12 PEX patients (12%) and 18 non-PEX patients (7.5%) suffered a stroke (*p* = 0.17, chi squared test).

During the follow-up period, 91 (26.9%) individuals in the overall sample (PEX and non-PEX cohorts, *n* = 338) died, 29 (29.3%) of whom were from the PEX group and 62 (25.9%) from the non-PEX group (*p* = 0.52; chi squared test).

## 4. Discussion

The prevalence of PEX syndrome in the overall study cohort was 7.57%, a figure that was very low compared to that reported in other studies where a higher prevalence of PEX was calculated in patients admitted for cataract surgery: 16.4% [8], 27.9% [9] and 17.4% [10]. However, the prevalence of PEX in our study is very similar to that observed in a Spanish geriatric population geographically related to ours and with a similar range of ages [11]. In this earlier study, the prevalence of PEX among those born within the study territory was 7.9% compared to 21.2% among those not born in the study region. Other authors also observed higher prevalence rates in different regions of Spain: 38.7% in Galicia [12], 25% in Tarragona [13], and 28% in Pontevedra [14]. Thus, the variability in the prevalence of this syndrome might well be explained by geographic, ethnic, and genetic variations.

Given the sample characteristics of this study (Spanish population, people over 80 years old), we decided to calculate CVR using the ERICE charts. New CVR prevention guidelines have recently been published, in which people up to 89 years of age can be evaluated [15]. Here, we found that the mean ERICE score in PEX patients (36.41% CVR) is higher than that in the non-PEX patients (33.72% CVR), although this difference was not statistically significant. This very high mean CVR in both groups of patients is striking, although it was expected given that the participants were elderly subjects with a mean age of 77.9 years old in the PEX group and 78.1 years old in the non-PEX group. Indeed, age is the strongest predictor of CVR in both men and women [7]. Risk stratification was similar in both groups, and although no significant differences were detected, there was a slightly higher proportion of patients with a very high CVR in the PEX group (68.4%) than in the non-PEX group (58.2%). To our knowledge, this is the first study to analyze the risk of developing cardiovascular disease in patients with PEX.

The proportion of patients with HBP was significantly higher in the PEX group (71.4%) than in the non-PEX group (58.6%), and we believe that this observation is fundamental when calculating the CVR. An association between PEX and HBP was observed previously in the Australian Blue Mountains Eye Study [16], from which it was concluded that PEX might reflect elastosis, which would favor an affectation of small-diameter vascular vessels and the development of vascular disease. An association between PEX and HBP has also been found elsewhere [17], although such an association has not always been described [18,19]. The relationship between HBP and PEX could have a histological background, with PEX syndrome associated with elastosis caused by excess synthesis and reduced degradation of the elastic microfibrils (including fibrillin-1). This phenomenon could have an effect on blood flow, since elastin is the main component of the extracellular matrix of arteries and arterioles. In fact, a number of authors have observed an association between vascular wall abnormalities and the presence of PEX material [20,21]. This anomaly of the vascular wall mainly affects the tunica intima, provoking an ensuing dysfunction of the endothelium. Indeed, an impaired endothelial vasodilatory response is often observed in PEX patients compared to those with normal blood vessels [22]. These findings suggest that the vascular dysregulation observed in the PEX syndrome could contribute to the pathophysiology of arterial diseases, including the development of HBP.

However, we did not find differences in the number of coronary events between PEX and non-PEX patients after a six-year follow-up, which is consistent with the absence of an association between ischemic heart disease and PEX in a Spanish population from Tarragona [13]. Nevertheless, this contrasts with the association between PEX and a history of ischemic heart disease found elsewhere [8,9,10,16,18], a discrepancy that may reflect different circumstances. Firstly, our study is longitudinal, and it analyzes the development of coronary events after following patients for six years. By contrast, in other published studies, the recording of ischemic heart disease was retrospective, using questionnaires, medical records, or electrocardiograms. Secondly, the participants in this study were not recruited from the population at large, but rather, their CVR factors were obtained from their medical records as recorded by their family doctors, and as such, they were receiving treatment to mitigate their elevated CVR. In other words, as their risk factors were being treated and monitored by their family physician, they suffered fewer cardiovascular events than the patients recruited from the population at large.

The prevalence of stroke was higher in the PEX group (12%) than in the non-PEX group (7.5%), although this difference was not statistically significant. This result is similar to that obtained from the population in Tarragona (Spain), in which a greater proportion of the patients with PEX had a history of cerebrovascular disease than the patients without PEX. However, when other CVR factors were entered into a logistic regression analysis, this association appeared not to be statistically significant [13]. Likewise, we did not observe significant differences in 6-year mortality between PEX and non-PEX patients, and although we are aware that this is a short period to assess mortality, it was not possible to carry out long-term mortality studies given the old age of our patients. However, no association between PEX syndrome and general all-cause mortality/overall mortality was evident in a 30-year mortality study [23], and likewise, no association was found between PEX syndrome and mortality due to cerebrovascular disease or heart disease [24,25,26].

This study has several limitations. First, the study was restricted to patients who underwent cataract surgery as opposed to studying the general population. In addition, the distribution of patients into the PEX and non-PEX cohorts was based on ophthalmological data from their medical records, and therefore, PEX may have been underdiagnosed. Finally, when calculating the CVR, the data were obtained from each patient’s medical records, such that the measurements were not obtained under the same conditions in all participants.

## 5. Conclusions

In conclusion, we found here that PEX was associated with HBP, although cardiovascular risk was not shown to be higher in this group. In addition, we did not find that patients with PEX experienced more cardiovascular events, nor have a higher mortality rate, over the six-year follow-up period studied.

## Figures and Tables

**Table 1 jcm-11-02153-t001:** Baseline characteristics and outcomes of the whole population.

Variables	Number of Patients *N* = 338		Mean	Standard Deviation
Sex	Female	Male		
	174 (51.5%)	164 (48.5%)		
Age			78.06	6.54
Smoker	32 (9.5%)			
Diabetes mellitus	73 (21.6%)			
HBP	210 (62.3%)			
Total cholesterol (mg/dL)			203.2	40.3
HDL cholesterol (mg/dL)			59.7	17.5
LDL cholesterol (mg/dL)			121.26	34.8
SBP (mm Hg)			136.1	12.9
DBP (mm Hg)			74.9	9.1
ERICE (%)			34.5	14.7
Cardiovascular event after 6-year follow-up	47 (13.4%)			

Abbreviations: HDL, high density lipoproteins; LDL, low density lipoproteins; HBP, high blood pressure; SBP, systolic blood pressure; DBP, diastolic blood pressure.

**Table 2 jcm-11-02153-t002:** Distribution of the PEX and Non-PEX cohorts by CVR factors.

Variables	Group				*p* Value
	PEX (*n* = 99)		Non-PEX (*n* = 239)		
Mean age (SD)	77.97 (6.42)		78.11 (6.62)		0.86
<60 years old	1 (1%)		1 (0.4%)		0.94
60–69 years old	10 (10.1%)		24 (10%)		
70–79 years old	47 (47.5%)		113 (47.3%)		
>80 years old	41 (41.4%)		101 (42.3%)		
Sex	Male (*n* (%))	Female (*n* (%))	Male (*n* (%))	Female (*n* (%))	0.19
	54 (54.5%)	45 (45.5%)	110 (46%)	129 (54%)	
Total cholesterol mg/dL (SD)	210.14 (40.4)		200.36 (40.03)		0.04
HDL cholesterol mg/dL (SD)	63.36 (18.7)		58.2 (16.8)		0.01
LDL cholesterol mg/dL (SD)	124.32 (35.4)		119.97 (34.6)		0.3
SBP (mmHg)	134.53 (11.4)		136.78 (13.4)		0.14
DBP (mmHg)	74.51 (8.9)		75.14 (9.1)		0.56
Smoker (*n* (%))	14 (14.1%)		18 (7.5%)		0.07
Diabetes mellitus (*n* (%))	20 (20.2%)		53 (22.2%)		0.77
HBP (*n* (%))	70 (71.4%)		140 (58.6%)		0.035

Abbreviations: PEX, pseudoexfoliation; CVR, cardiovascular risk; SD, standard deviation; HDL, high density lipoproteins; LDL, low density lipoproteins; SBP, systolic blood pressure; DBP, diastolic blood pressure; HBP, high blood pressure.

**Table 3 jcm-11-02153-t003:** Calculation of the CVR according to ERICE in the PEX and non-PEX cohorts.

	ERICE						Mean
	Low <5%	Mild 5–9%	Moderate 10–14%	Moderate-High 15–19%	High 20–29%	Very High >30%	
PEX	0 (0%)	0 (0%)	2 (2.0%)	11 (11.2%)	18 (18.4%)	68 (68.4%)	36.41%
Non- PEX	0 (0%)	9 (3.8%)	4 (1.7%)	44 (18.4%)	43 (18%)	139 (58.2%)	33.72%
*p* value	0.13						0.13

The distribution of the patients in each ERICE category and the mean CVR are shown and there were no differences in the mean CVR between the PEX and non-PEX patients (the values represent the number of patients, expressed as a percentage). A comparison of the proportions of the different ERICE categories was conducted using a chi-squared test. The means were compared using Student’s *t*-test. PEX: pseudoexfoliation. CVR: cardiovascular risk.

**Table 4 jcm-11-02153-t004:** Analysis of the relationship of PEX with HBP.

	Univariate Analysis					Multivariate Analysis	
	With HBP *n* = 211		Without HBP *n* = 127		*p* value	Odds ratio (OR)	Confidence interval (CI)
PEX	PEX	Non-PEX	PEX	Non-PEX	0.03	1.79	1.06–3.1
	71 (33.6%)	140 (66.3%)	28 (22%)	99 (78%)			

HBP is associated with the presence of PEX. The values indicate the number of patients (percentage). PEX, pseudoexfoliation; HBP, high blood pressure.

## Data Availability

Not applicable.

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
