# Peer review of "Comparison of Cardiovascular Risk and Events among Spanish Patients with and without Ocular Pseudoexfoliation"

_jcm, 2022, doi:10.3390/jcm11082153_

Round 1

Reviewer 1 Report

Dear Editor,
I have read with extreme interest the article by Aristimuño et al. entitled "Comparison of cardiovascular risk and events among Spanish patients with and without ocular pseudoexfoliation". The paper was aimed at assessing the relationship between ocular pseudoexfoliation (PEX) and cardiovascular disease (CVD) and in particular whether PEX is an independent risk factor for CVDs. in order to determine that, patients were longitudinally studied and a global cardiovascular risk score (CVR) was assessed by means of the ERICE score (https://www.revespcardiol.org/en-the-erice-score-new-native-cardiovascular-articulo-S1885585714002448) with and without taking into account the PEX contribution.
Recent studies on the PEX has shown that PEX syndrome is an age-related systemic disease that mainly affects the anterior structures of the eye. Despite a worldwide distribution, reported incidence and prevalence of this syndrome vary widely between ethnicities and geographical areas. The exfoliative material is composed mainly of abnormal cross-linked fibrils that accumulate progressively in some organs such as the heart, blood vessels, lungs or meninges, and particularly in the anterior structures of the eye.
The exact pathophysiological process still remains unclear but the association of genetic and environmental factors are thought to play a role in the development and progressive extracellular accumulation of exfoliative material. In addition to that,  LOXL1 gene polymorphisms, responsible for metabolism of some components of elastic fibers and extracellular matrix, and increased natural exposure to ambient ultraviolet or caffeine consumption have been associated with the PEX syndrome.

The main finding of this paper by Aristimuño et al is a statistically significant association of high blood pressure (HBP) in the PEX group (71.4%) than in non-PEX group (58.6%), fundamental feature for the assessment of global CVR. This is probably due to the underlying elastosis that affects the small-diameter vascular vessels.

The paper by Aristimuño et al. demonstrates how collagen disorders (traditionally associated to vascular disorders like the case of Marfan Syndrome, Loeys-Dietz Syndromes, Ehlers-Danlos Syndorme etc) can actually contribute to ishemic heart disease in a more subtle way than other factors suche es tChol, chol-LDL etc etc.

Under this light this paper deserves attention.

Author Response

Dear reviewer,

Thank you for your kind comments. Below, we attached file with the corrections suggested by the other reviewers.

Kind regards.

Reviewer 2 Report

Dear Authors!

Thank you for submitting your manuscript to Journal of Clinical Medicine. It is an honour for me to review your article.

Your research is a well-planned cohort study investigating cardiovascular risk and events in patients with and without ocular pseudoexfoliation. It has several strenghts including nice sample size, long follow-up, surrogate as well as clinical endpoints.

Below I have typed my comments and suggestions regarding your manuscript.

  1. There were 1,426 cataract interventions (June - December 2016), and you have finally enrolled 99 PEX and 239 non-PEX patients. What about follow-up? Did you maintained 100% follow-up after 6 years of your study? Could you present detailed flow-chart of enrolled patients, please?
  2. Have you tried to explore the impact of presence of unilateral vs bilateral PEX on outcomes as it may represent the severity of disease?
  3. Have you utilized the proper ERICE charts to each patient - diabetic vs non-diabetic and hypertensive vs non-hypertensive? I suggest you to add such statement in Methods section.
  4. While in Results ("When the CVR was calculated using the ERICE charts, no statistical differences between PEX and non-PEX patients were detected") or Discussion sections ("the mean ERICE score in PEX patients (36.41% CVR) is higher 194 than that in the non-PEX patients (33.72% CVR), although this difference was not statistically significant.") you have highlighted that differences in CVR between cohorts were not significant, your conclusions seem to be other in the Abstract ("PEX was associated with an increased CVR in our 26 cohort of patients") or Conclusions section ("we found here that PEX was associated with an increased CVR"). Could you explain it or make it more consistent thorough the manuscript?

I am looking forward for your responses!

Author Response

Dear reviewer,

Thank you very much for your interesting contributions to our manuscript. Below, we will show our responses to your suggestions.

  1. We followed-up during six years all the patients enrolled (n=338): During the six years of follow-up, 47 (13,4%) patients in the overall sample (n = 338) suffered a cardiovascular event, of which 17 patients suffered acute ischemic heart disease (5%) and 30 had an acute stroke (8,8%). A detailed flow-chart of enrolled patients is attached.
  2. We find it very interesting to compare the presence of unilateral vs bilateral PEX and analyze cardiovascular events. In eyes with clinically visible unilateral PEX, the presence of PEX material has been evidenced in pathological samples of contralateral eyes, which suggests that PEX often is bilateral, although clinically it is not visible. We think that the presence of bilateral PEX would be associated with greater accumulation of fibrinoid material in the blood vessels, which would mean greater deregulation of the vascular system.
  3. Proper ERICE chart was used for each patient (please see such statement in Methods section)
  4. In conclusion, we found here that PEX was associated with HBP, although cardiovascular risk was not shown to be higher in this group. In addition, we did not find that patients with PEX experienced more cardiovascular events or have a higher mortality rate over the 6 year follow-up studied.

Reviewer 3 Report

This is a very interesting analysis associating PEX with CV risk and events. While the manuscript is well-written, the methodology may be optimized and the authors seem to jump to the wrong conclusions:
•    The methodology of matching is unclear to me: did the authors use every matching patient or limit the maximum number of matches? „2-3“ is just not exact enough. Furthermore, the authors may have found an increased proportion of hypertensive patients in the PEX group just by matching wrongly. I suggest to use propensity score matching. Why did the authors choose to use matching abyways?
•    A study flowchart is needed.
•    Authors should use SCORE2 as well, as recommended by ESC guidelines.
•    A table showing baseline characteristics and outcome of the whole population is needed (not just matched patients).
•    Lastly, but most importandly: a p value > 0.05 means no significance = no difference! The authors cannot argue changes that are not there. There was no significant difference in cardiovascular risk or cardiovascular events, according to the statistics performed by the authors. The fiscussion has to be adapted adequately.

Author Response

Dear reviewer,

Thank you very much for your interesting contributions to our manuscript. Below, we will show our responses to your suggestions.

1. Methodology of matching: the two cohorts of this study were obtained by random sampling from the list of patients who were cited in the post-operative evaluation carried out the day after the cataract intervention during the period from June to December 2014. The cohorts PEX and non-PEX were obtained from the patient´s ophthalmological medical records: if the medical record specified that the patient had PEX, that patient would be part of the PEX cohort. If no mention of PEX was made in the medical record, that patient would be part of the non-PEX cohort.

PEX cohort formation: all patients whose medical history included signs of PEX in one or both eyes were included in the PEX cohort.

Non-PEX cohort formation: was formed by patients whose medical history did not mention the presence of signs of PEX. For each PEX patient included, 2-3 consecutive age (± 5 years) and gender matched non-PEX patients were also included. The selection of non-PEX patients was carried out randomly, following the order of the list of patients cited in the postoperative consultation. If in the same day list there were 3 or more non-PEX patients of the same age and sex range as the reference PEX patient, 3 consecutive non-PEX patients were chosen. If in the same day list there were only 2 non-PEX patients of the same age and sex range as the reference PEX patient, 2 non-PEX patients were chosen for each PEX patient

2. Study flowchart: please see the attachment

3. Most existing risk calculation systems have been developed from cohorts consisting mainly of middle-aged people, leaving older people underrepresented. The SCORE function focuses on middle-aged people and is only recommended for use within the age group 45-65 years. In addition, it does not take into account the possible effects of pharmacological treatment of risk factors in the study population, so it underestimates the actual cardiovascular risk. According to the joint working group's guidance Fourth Joint Task Force, it is important to develop a European cardiovascular disease risk score for specific geographical areas and include elderly individuals to enable more accurate identification of asymptomatic individuals at high risk of cardiovascular disease. The ERICE-charts estimate the absolute 10-year cardiovascular risk for a first fatal or non-fatal cardiovascular event, and the risk factors included are age, smoking, diabetes mellitus, systolic blood pressure and total colesterol. Moreover, the ERICE data are also characterized by including sections for people aged 70-79 or over 80 years of age, and it takes into account the use of drugs that modify specific risk factors (e.g., anti-high blood pressure treatment). Consequently, it is necessary to start paying more attention to the estimation of cardiovascular risk in very elderly individuals. Individuals over the age of 80 have been shown to have a beneficial effect with preventive interventions such as treatment with antihypertensive drugs if they have hypertension. Therefore, given the sample characteristics of this study (Spanish population, people over 80 years old) we have decided to calculate the cardiovascular risk using the ERICE charts.

4. A table showing baseline characteristics and outcome of the whole population: please see the attachment

5. In conclusion, we found here that PEX was associated with HBP, although cardiovascular risk was not shown to be higher in this group. In addition, we did not find that patients with PEX experienced more cardiovascular events or have a higher mortality rate over the 6 year follow-up studied.

Round 2

Reviewer 2 Report

Dear Authors, thank you for your responses!

Author Response

Thank you very much.

Reviewer 3 Report

Thank you for the improvements.

However, in my opinion the matching is still inappropriate as inbalanced groups were created. I do not understand that the authors matched based on the day of surgery. I strongly recommend to either use propensity score matching (or similar methods) or remove matching from the analysis as it is not necessary when using COX regression.

Furthermore, I want to emphasize the importance of widely accepted cardiovascular risk scores. I want to clarify that I meant the SCORE2 and not SCORE. This risk score is endorsed by the 2021 ESC CV prevention guidelines (Visseren, F. L. J., et al. (2021). "2021 ESC Guidelines on cardiovascular disease prevention in clinical practice." Eur Heart J 42(34): 3227-3337). Together with the SCORE2, people up to 89 years of age can be evaluated.

Author Response

Dear reviewer,

  1. The purpose of including non-PEX patients on the same day as the PEX patients was to obtain informed consent when the patients came for post-operative evaluation. All the patients were enrolled  the day after surgery and the informed consent was signed that day. If we had done it differently, patients would have to come to the hospital 2 times. The reason for including more than 1 non-PEX patient was to increase the sample size. The cohorts formation was matched (2-3 non PEX for each PEX patient), but the analysis was not matched. Matching the patients we aimed to control sex and age in both groups. However, we are aware that there are better ways to obtain the cohorts. If you agree, we will remove matching from methods and from analysis as it does not influence the results.
  2. We have introduced a new paragraph in Discussion: Given the sample characteristics of this study (Spanish population, people over 80 years old) we decided to calculate the cardiovascular risk using the ERICE charts. New cardiovascular risk prevention guidelines have recently been published, by which people up to 89 years of age can be evaluated